# A High Dietary Incorporation Level of *Chlorella vulgaris* Improves the Nutritional Value of Pork Fat without Impairing the Performance of Finishing Pigs

**DOI:** 10.3390/ani10122384

**Published:** 2020-12-12

**Authors:** Diogo Coelho, José Pestana, João M. Almeida, Cristina M. Alfaia, Carlos M. G. A. Fontes, Olga Moreira, José A. M. Prates

**Affiliations:** 1CIISA-Centro de Investigação Interdisciplinar em Sanidade Animal, Faculdade de Medicina Veterinária, Universidade de Lisboa, 1300-477 Lisboa, Portugal; diogocoelho@fmv.ulisboa.pt (D.C.); jpestana@fmv.ulisboa.pt (J.P.); cpmateus@fmv.ulisboa.pt (C.M.A.); cafontes@fmv.ulisboa.pt (C.M.G.A.F.); 2INIAV-Instituto Nacional de Investigação Agrária e Veterinária, Fonte Boa, 2005-048 Vale de Santarém, Portugal; joaoalmeida@iniav.pt (J.M.A.); olga.moreira@iniav.pt (O.M.)

**Keywords:** *Chlorella vulgaris*, CAZymes, finishing pigs, growth performance, pork quality, fat composition

## Abstract

**Simple Summary:**

Pork is one of the most consumed meats worldwide but its production and quality are facing significant challenges, including feeding sustainability and the unhealthy image of fat. In fact, corn, and soybean, the two main conventional feedstuffs for pig production, are in unsustainable competition with the human food supply and biofuel industry. Moreover, the nutritional value of pork lipids is small due to their low contents of the beneficial *n*-3 polyunsaturated fatty acids and lipid-soluble antioxidants. The inclusion of microalgae in pig diets represents a promising approach for the development of sustainable pork production and the improvement of its quality. The current study aimed to investigate the impact of *Chlorella vulgaris* as ingredient (5% in the diet), alone and in combination with carbohydrases, on growth performance, carcass characteristics and pork quality traits in finishing pigs. Our data indicate that the use of 5% *C. vulgaris* in finishing pig diets does not impair animal growth and ameliorates the nutritional value of pork. Therefore, *C. vulgaris* could be used advantageously as an alternative sustainable ingredient in swine feeding.

**Abstract:**

The influence of a high inclusion level of *Chlorella vulgaris*, individually and supplemented with two carbohydrase mixtures, in finishing pig diets was assessed on zootechnical performance, carcass characteristics, pork quality traits and nutritional value of pork fat. Forty crossbred entire male pigs, sons of Large White × Landrace sows crossed with Pietrain boars, with an initial live weight of 59.1 ± 5.69 kg were used in this trial. Swines were randomly assigned to one of four dietary treatments (*n* = 10): cereal and soybean meal-based diet (control), control diet with 5% *C. vulgaris* (CV), CV diet supplemented with 0.005% Rovabio^®^ Excel AP (CV + R) and CV diet supplemented with 0.01% of a four-CAZyme mixture (CV + M). Animals were slaughtered, after the finishing period, with a BW of 101 ± 1.9 kg. Growth performance, carcass characteristics and meat quality traits were not influenced (*p* > 0.05) by the incorporation of *C. vulgaris* in the diets. However, the inclusion of the microalga in finishing pig diets increased some lipid-soluble antioxidant pigments and *n*-3 PUFA, and decreased the *n*-6:*n*-3 ratio of fatty acids, thus ameliorating the nutritional value of pork fat. Moreover, the supplementation of diets with the carbohydrase mixtures did not change (*p* > 0.05) neither animal performance nor meat quality traits, indicating their inefficacy in the increase of digestive utilization of *C. vulgaris* by pigs under these experimental conditions. It is concluded that the use of *C. vulgaris* in finishing pig diets, at this high incorporation level, improves the nutritional value of pork fat without compromising pig performance.

## 1. Introduction

Pork industry is currently facing the big challenges of feeding sustainability and the unhealthy image of fat. In fact, pork production is about 38% of the total amount of meat produced in the world, being is the most commonly consumed meat in different European, American, and Asian countries [1]. Moreover, the combination between rise of global population and the increase in income, will double the overall demand for animal-derived products by 2050, including pork [2]. The increased demand for these products will necessarily bring dramatic consequences in terms of sustainability, as cereal grains and soybean food crops are the two main conventional feedstuffs for animal feeding [3]. Therefore, alternative feed ingredients are needed to sustain animal agriculture and human food security [4,5].

In addition, pork is frequently considered unhealthy due to the lower proportions of polyunsaturated fatty acids (PUFA) and lipid-soluble antioxidant vitamins, and higher percentages of saturated fatty acids (SFA) [6]. However, it is well established that pig diet provides an effective approach for altering the fat composition of pork, thereby modifying the impact of human dietary fat intake from pork [7]. Functionally, the most important *n*-3 fatty acids are eicosapentaenoic acid (EPA, 20:5*n*-3) and docosahexaenoic acid (DHA, 22:6*n*-3), although the roles for docosapentaenoic acid (DPA, 22:5*n*-3) are now also emerging [8]. Lipid-soluble antioxidant vitamins comprise vitamin E homologues (tocopherols and tocotrienols) and vitamin A and its precursors (some carotenoids, including β-carotene). In general, the intakes of EPA and DHA are typically small and much lower than the recommended values [9]. This fact raised substantial interest in food enrichment with EPA and DHA, by using feed ingredients from marine origin in animal nutrition.

Microalgae, an important aquatic resource, could be a good sustainable alternative to conventional feedstuffs, since they have similar nutritional compositions [10]. *Chlorella vulgaris* is a freshwater eukaryotic green microalga. This microalga, one of the most cultivated microalgae worldwide, is known for its high biomass productivity, relative ease of cultivation and a balanced nutritional composition, making it an attractive alternative for monogastric diets [11]. In particular, regarding fatty acid profile, *C. vulgaris* displays a high percentage of SFA, mainly myristic acid (14:0), palmitic acid (16:0) and stearic acid (18:0). In addition, *C. vulgaris* presents an interesting content in some *n*-6 PUFA (18:2*n*-6 and 18:3*n*-6) and α-linolenic acid (18:3*n*-3), but much less quantity of EPA and DHA [12]. However, *C. vulgaris* cell wall is composed by a diverse and complex matrix of cross-linked insoluble carbohydrates [13]. Thus, the incorporation of *C. vulgaris* in monogastric diets could be a problem since the recalcitrant cell wall is largely indigestible, impairing the bioavailability of its valuable nutrients [14].

Exogenous carbohydrate-active enzymes (CAZymes) are now completely accepted as feed supplements for monogastric livestock species to improve feed nutritive value and enhance animal performance and health [15]. In addition to cereal cell walls, several in vitro studies demonstrated the ability of CAZymes to degrade microalgae cell walls [16,17,18,19]. Recently, Coelho et al. [20] described a four-CAZyme mixture, composed by an exo-β-glucosaminidase, an alginate lyase, a peptidoglycan N-acetylmuramic acid deacetylase and a lysozyme, that was shown to disrupt microalgae *C. vulgaris* cell walls to a significant extent, in in vitro assays, enabling the release of trapped nutrients with important nutritional value.

Therefore, the supplementation with the four-CAZyme mixture mentioned above could enable the incorporation of *C. vulgaris* in monogastric diets, at high incorporation levels (>2% in diet), without impairing animal performance and health. In line with this, the aim of this study was to assess how the dietary incorporation of *C. vulgaris* at a 5% high level, supplemented or not with two exogenous CAZyme mixtures (the commercially available Rovabio^®^ Excel (ADISSEO, Antony, France) AP and the four-CAZyme mixture described by Coelho et al. [20]), influences finishing pigs’ performance, carcass characteristics, and pork quality traits.

## 2. Materials and Methods

### 2.1. Production of Recombinant Four-CAZyme Mixture

The genes encoding the four recombinant CAZymes that compose the mixture (exo-β-glucosaminidase, alginate lyase, peptidoglycan N-acetylmuramic acid deacetylase and lysozyme) were cloned according to Coelho et al. [20]. Briefly, BL21 *Escherichia coli* cells were transformed with the generated recombinant plasmids and were grown on Luria–Bertani media, at 37 °C under agitation (190 rpm) to mid exponential phase (absorbance was measured at λ = 595 nm as being 0.4–0.6). Isopropyl β-d-thiogalactoside was added to a final concentration of 1 mM in order to induce recombinant gene expression. Cells were incubated overnight at 19 °C with agitation (140 rpm). After induction, the culture media was centrifuged and the protein extracts were prepared by ultrasonication, followed by centrifugation and freeze dried. The four-CAZyme protein extracts were mixed in equal weight proportions at a final level of 0.01%.

### 2.2. Animal Care, Experimental Design and Experimental Diets

The trial was conducted at the facilities of Unidade de Investigação em Produção Animal (Instituto Nacional de Investigação Agrária e Veterinária (UEISPA-INIAV, Santarém). The experimental procedures were reviewed by the Ethics Commission of the Centro de Investigação Interdisciplinar em Sanidade Animal/Faculdade de Medicina Veterinária (CIISA/FMV) and approved by the Animal Care Committee of the National Veterinary Authority (Direção-Geral de Alimentação e Veterinária), following the appropriated European Union guidelines (2010/63/EU Directive). The staff members involved in animal trial hold license for conducting experiments on live animals from the Portuguese Veterinary Services.

Forty crossbred entire male pigs, sons of Large White × Landrace sows crossed with Pietrain boars, were obtained from a commercial farm. Before the beginning of the trial, pigs were submitted to an adaptation period of one week. Then, pigs with an initial weight of 59.1 ± 5.69 kg were randomly distributed into 10 pens with 4 animals in each pen (7.8 m^2^). Pens were equipped with one stainless steel nipple and four creep feeders separated by a system of gates, thus allowing individual feed intake control. The 4 experimental diets were randomly assigned to the four animals within each pen, with each animal in each pen receiving a different diet, thus being the pig the experimental unit. Pigs had access to feed and water ad libitum. The experimental diets were: cereal and soybean meal-based diet (Control), control diet with 5% of *C. vulgaris* supplied by Allmicroalgae (Natural Products, Portugal) (CV), control diet with 5% of *C. vulgaris* supplemented with 0.005% of Rovabio^®^ Excel AP (Adisseo, Antony, France) (CV + R), and control diet with 5% of *C. vulgaris* supplemented with 0.01% of four-CAZyme mixture (CV + M).

The ingredient composition of the experimental diets is described in Table 1, and their chemical composition is presented in detail in Table 2. For further information on the feed analysis see details below.

### 2.3. Animal Performance, Slaughter, and Sampling

During the experiment, supplied feed and refusals were recorded daily, whereas pig were weighed weekly just before feeding, with the purpose of calculate average daily feed intake (ADFI), average daily weight gain (ADG), feed conversion ratio (FCR) and gain:feed ratio (G:F). Food was withdrawn from animals 17–19 h before slaughter. Animals were slaughtered at a BW of 101 ± 1.9 kg, after a trial period of 41 ± 7.8 days, at the Unidade de Investigação em Produção Animal experimental slaughterhouse (Santarém, Portugal), with electrical stunning followed by exsanguination. The hot carcass weight (HCW) was measured in order to calculate carcass yield. Perirenal and mesenteric fat depot was removed and weighed. Longissimus lumborum muscle was collected from the right carcass side between the third and fifth lumbar vertebras, minced, immediately vacuum packed and stored at −20 °C, to assess meat quality, and at −80 °C, for meat oxidative stability determinations.

At 24 h post mortem, backfat thickness was measured in the left side of carcass at the last rib position (P2) (the most representative location), last lumbar vertebra (L6) and second sacral vertebra (S2), using a calibrated engineering calliper (150 mm Electronic Digital Vernier Calliper CE ROHS) as described by Frederick [21]. The loin was excised from the left side of carcass, between the last cervical and L6 lumbar vertebras, weighted and sliced into 2.5-cm-thick chops for sensory evaluation, shear force measurements, color and drip loss determinations. Chops were vacuum packed, frozen and stored at −20 °C until sensory analysis and shear force measurements.

### 2.4. Microalga and Experimental Diets Analyses

Experimental diets were collected three times during the entire trial. The Association of Official Analytical Chemists (AOAC) [22] methods were used to determine the proximal composition of *C. vulgaris* microalga and experimental diets. Samples were dried at 103 °C until reach constant weight to determine dry matter (DM). Crude protein of samples was calculated through the determination of the nitrogen content (N) by the Kjeldahl method using the factor 6.25 × N following the method 954.01 [22]. Ash and starch contents of samples were determined according to the method 942.05 [22] and Clegg [23] procedure, respectively. Crude fat of samples was determined after automatic Soxhlet extraction with petroleum ether (Gerhardt Analytical Systems, Königswinter, Germany). Crude fiber, acid detergent fiber (ADF) and neutral detergent fiber (NDF) were determined by the method 989.03 [22]. Metabolizable energy (ME) was calculated according to Noblet et al. [24].

The amino acid composition of *C. vulgaris* and experimental diets was determined according to the method 994.12 [25] and quantified by High-Performance Liquid Chromatography (HPLC) (Agilent 1100, Agilent Technologies, Avondale, PA, USA), as described by Henderson et al. [26]. The fatty acid methyl esters (FAME) profile of *C. vulgaris* and experimental diets were analyzed by one-step extraction and acid transesterification, followed by gas chromatography using heneicosaenoic acid (21:0) methyl ester as the internal standard [27].

The diterpene profile of *C. vulgaris* and experimental diets was analyzed by direct saponification, using a single *n*-hexane extraction followed by HPLC analysis [28]. The determination of pigments in *C. vulgaris* and experimental diets was performed according to Teimouri et al. [29], with slight modifications as described in Pestana et al. [30]. The quantification of pigments in *C. vulgaris* and experimental diet samples were performed according to Hynstova et al. [31].

### 2.5. Meat Quality Traits

The pH and temperature of longissimus lumborum muscle were measured in the right carcass side at 45 min and 24 h post mortem using a pH meter equipped with a penetrating electrode (HI8424, Hanna Instruments, Woonsocket, RI, USA). Meat color was measured on the cut surface of longissimus lumborum section, 24 h post mortem, using a colorimeter (Minolta CR-400, Konica Minolta, Tokyo, Japan) with the illuminant D65, at an observer angle of 2° and 1 cm diameter of measurement area. Three measurements on different locations per sample were recorded following the CIE color convention L* (lightness), a* (redness) and b* (yellowness) system after 1 h of blooming at 4 °C [32].

Drip loss of fresh longissimus lumborum muscle was performed according to Hope-Jones et al. [33]. The amount of drip measured between 24 h and 144 h post mortem was expressed as a percentage of the initial mass of the sample, and calculated through the difference between the sample mass at the beginning and end of the storage period.

### 2.6. Cooking Loss and Shear Force Measurements 

Meat cooking loss and shear force were determined according to the procedure adapted from Honikel [34] and Oillic et al. [35]. Frozen meat samples were thawed at 2 ± 1 °C overnight, weighed and cooked in a water bath at 80 ± 0.5 °C until reaching the temperature of 75 ± 0.5 °C in the geometric center, using an internal thermocouple (Thermometer Omega RDXL4SD, Manchester, NH, USA). The samples were cooled for 20 h (2 ± 1 °C), weighed in order to calculate the cooking loss, and longitudinally cut in the fiber axis parallel to muscle fiber direction into 8 to 12 cores, with a 1-cm^2^ cross-section area for shear force determinations. Cooking loss, expressed as percentage, was calculated by the difference of the weights before and after cooking divided by the initial weight of the sample [36].

The Warner–Bratzler shear force (WBSF) was measured in a texture analyzer (TA-XT Plus texture analyzer; Stable Micro Systems, Surrey, UK) with a Warner–Bratzler shear device with a 30-kg compression load cell, trigger force was 25 g and crosshead speed during pre-test, test and post-test set were 5.0, 2.0, and 10.0 mm/s, respectively. Force and distance were recorded at 200 points/s and analyzed with the Version 6.1.16 of Exponent software (Stable Micro Systems, Surrey, UK). The value of the peak shear force of cores from each sample was determined and averaged to obtain a single WBSF value per sample.

### 2.7. Trained Sensory Panel Analysis 

A trained sensory panel with five sessions was used to evaluate meat sensory characteristics. The eleven panelists were selected and trained according to Cross et al. [37]. For each session, meat samples were thawed at 2 ± 1 °C overnight and cooked at 170 ± 5 °C in a Ceramic Contact Grill 1.6 kW (UNOX Catering Equipment, Padova, Italy) with an internal thermocouple in each sample (Thermometer Omega RDXL4SD, Manchester, NH, USA) until reached 71 °C in the geometric center. After 10 min of stabilization at 40 °C, the sample was trimmed of the six external surfaces, included connective tissue, cut into 1 × 1 × 1 cm subsamples and maintained, on individual covered plates, in an incubator at 40 °C until tasting (no longer than 30 min) [38]. Samples were randomly distributed across sections and the attributes evaluated were juiciness, tenderness, flavor intensity, off-flavor, flavor acceptability, and overall acceptability. These attributes were classified on a grading scale from 1 (extremely dry, tough, soft, weak, or unacceptable) to 8 (extremely juicy, tender, strong, positive and positive), with the exception of off-flavor quantified from 0 (absence) or 1 (presence) [39].

### 2.8. Determination of Total Cholesterol and Diterpene Profile in Meat

The simultaneous quantification of total cholesterol, β-carotene and vitamin E homologues (tocopherols and tocotrienols) in longissimus lumborum samples was performed, in duplicate, as previously described by Prates et al. [28]. Muscle samples were submitted to a saponification reaction in a water bath at 80 °C for 15 min under agitation. Afterwards, the diterpenes were extracted with *n*-hexane and analyzed by HPLC system (Agilent 1100 Series, Agilent Technologies Inc., Palo Alto, CA, USA), using a normal-phase silica column (Zorbax RX-Sil, 250 mm × 4.6 mm i.d., 5 μm particle size, Agilent Technologies Inc, Santa Clara, CA, USA). The HPLC analysis was performed using UV-visible photodiode array detector for cholesterol (λ = 202 nm) and β-carotene (λ = 450 nm) coupled to fluorescence detector for tocopherols and tocotrienols (excitation λ = 295 nm and emission λ = 325 nm). Standard curves of peak area *versus* concentration was used to quantify total cholesterol, β-carotene and vitamin E homologues contents in meat samples.

### 2.9. Determination of Pigments in Meat

The contents of chlorophyll *a*, chlorophyll *b* and total carotenoids were measured according to the procedure of Teimouri et al. [29] modified by Pestana et al. [30]. One g of each sample was incubated overnight with 10 mL of acetone (Merck KGaA, 249 Darmstadt, Germany) under agitation at room temperature in absence of light. Then, samples were centrifuged at 3345× *g* for 5 min and the absorbance was measured in the supernatants using a UV–VIS spectrophotometer (Ultrospec 3100 pro, Amersham Biosciences, Little Chalfont, UK). The results were calculated according to Hynstova et al. [31], as described above for *C. vulgaris* microalga and experimental diets.

### 2.10. Determination of Total Lipid Content and Fatty Acid Composition

Longissimus lumborum muscle samples were lyophilized (−60 °C and 2.0 hPa) using a lyophilizator Edwards Modulyo (Edwards High Vacuum International, Crawley, UK) for total lipids determination according to Folch et al. [40]. Total lipid content was determined gravimetrically, in duplicate, by weighing the fat residue obtained after solvent evaporation. Then, the fat residue was re-suspended in dry toluene and submitted to sequential alkaline and acid transesterification procedure at 50 °C for 30 and 10 min, respectively, to convert fatty acids into FAME [41]. FAME were separated through gas chromatography (HP7890A Hewlett-Packard, Avondale, PA, USA) comprising a Supelcowax^®^ 10 capillary column (30 m × 0.20 mm internal diameter, 0.20 μm film thickness; Supelco, Bellefonte, PA, USA) and a flame ionization detector as described by Madeira et al. [42]. For FAME identification, a reference standard (FAME mix 37 components, Supelco Inc, Bellefonte, PA, USA) was used and confirmed by gas chromatography with a mass spectrometry detector using a GC-MS QP2010-Plus (Shimadzu, Kyoto, Japan). FAME were quantified by the internal standard method using heneicosanoic acid (21:0) methyl ester as internal standard. The fatty acids identified were expressed as percentage of total fatty acids.

### 2.11. Determination of Meat Lipid Oxidation

The extent of meat lipid oxidation was evaluated at day 0, 4 and 8 post mortem (storage at 4 °C), by quantifying thiobarbituric acid reactive substances (TBARS), following the spectrophotometric method described by Grau et al. [43]. TBARS values were calculated, in duplicate, from a standard curve constructed with 1,1,3,3-tetraethoxypropane (Fluka, Neu Ulm, Germany), as a precursor of malonaldehyde, and the results were presented as mg of malonaldehyde per kg of meat [42]. In addition, lipid peroxidation levels in meat were also measured by the concentration of TBARS, after chemical oxidation through a ferrous-hydrogen peroxide system, as described by Mercier et al. [44]. The TBARS were quantified after 0, 30, 120, and 300 min of oxidation induction following the method described above.

### 2.12. Statistical Analysis

All data were checked for normal distribution and variance homogeneity. Data were analyzed by analysis of variance using the PROC GLM of SAS software package (version 9.4; SAS Institute Inc., Cary, NC, USA) and measurements over time analyzed with PROC MIXED of SAS. The statistical model considered the dietary treatment the fixed effect and the pig the experimental unit. Least square means for multiple comparisons were generated using the PDIFF option adjusted with the Tukey–Kramer method. The significance level was set at *p* < 0.05.

## 3. Results and Discussion

### 3.1. Feed Intake, Growth Performance and Carcass Characteristics of Pigs

Data on feed intake, growth performance and carcass traits of finishing pigs are shown in Table 3. Growth performance variables had no significant differences among animals fed with different experimental diets (*p* > 0.05). The average values of ADG, ADFI, and FCR were 1.02 kg, 2.62 kg, and 2.59 kg, respectively. No significant differences in carcass characteristics were obtained among the experimental groups (*p* > 0.05), with the exception of perirenal fat (*p* = 0.026). The control group displayed a higher value of perirenal fat than the group fed with the *C. vulgaris* diet (+34%).

We assessed, for the first time, the impact of a high dietary level (>2% in diet) of *C. vulgaris*, individually and combined with two exogenous CAZymes, on pig performance. In fact, some studies reported the use of *C. vulgaris* in pig diets but at much lower levels (1% in the diet or lower), compared with the 5% incorporated in the current trial [45,46,47]. Baňoch et al. [46] investigated the effect of a very low level (0.0002%) of incorporation of *C. vulgaris* in female pigs, with an initial weight of 30 kg, and found no significant differences in ADG, HCW, lean muscle thickness and backfat thickness. Later, Furbeyre and colleagues [47] showed no significant effects on ADG, ADFI, and FCR, by using 1% of *C. vulgaris* in weaned piglet diets, with an initial weight of 9.1 kg, during 14 days. In another study, the same authors assessed the effect of oral supplementation with *C. vulgaris* (385 mg/kg BW) on growth and digestive health of weaning piglets and also found no significant changes in ADG, ADFI and G:F [48]. In addition, a study conducted in growing pigs, with an initial weight of 26.6 kg and *C. vulgaris* incorporation of 0.1% and 0.2% in the diet, described an increase of ADG with the lower dietary level without significant variations in ADFI and G:F [45]. In the present study, no significant effects on zootechnical parameters and carcass characteristics were obtained, which indicates that dietary incorporation of 5% *C. vulgaris* does not compromise the productive parameters of finishing pigs. Moreover, the dietary supplementation with exogenous carbohydrases, aiming at improving *C. vulgaris* digestibility by finishing pigs, does not seem to be necessary at this high incorporation level.

### 3.2. Pork Quality Traits and Sensory Evaluation

Data concerning the effect of experimental diets on quality traits of longissimus lumborum muscle from finishing pigs are presented on Table 4. Experimental treatments had no significant effect on temperature 45 min post mortem, pH 45 min and 24 h post mortem, color parameters, WBSF and cooking loss (*p* > 0.05). Table 5 summarizes the trained panel scores obtained for pork. No significant differences were obtained among experimental diets for the several items evaluated by the trained sensory panel (*p* > 0.05).

Similar results for meat quality traits were reported by Baňoch et al. [46], who found that a 0.0002% level of incorporation of *C. vulgaris* in pig diets had no significant effect on color, pH, cooking loss and drip loss of pork. Here, the dietary incorporation of 5% *C. vulgaris* did not change pork quality traits and sensory parameters, which is very important for the consumer acceptance of this meat. By contrast, Oh et al. [49] observed an increase of b*, pH and shear force in breast meat, and an increase of L* and b* in leg meat, of male Pekin ducks fed with 0.1–0.2% *C. vulgaris* during 42 days. Therefore, pork quality traits seem to be less sensitive to the dietary inclusion of *C. vulgaris* than poultry meat characteristics, although both are meats-derived from monogastric animals. Finally, it was also indicated here that the dietary use of CAZyme mixtures does not affect pork quality characteristics.

### 3.3. Vitamin E Profile and Pigments of Pork

The effect of experimental diets on vitamin E profile and pigments of longissimus lumborum muscle from finishing pigs is shown in Table 6. Experimental diets did not contribute to significant differences on the diterpene profile (*p* > 0.05). Regarding pigments analysis, β-carotene was not detected in any group fed with experimental diets, and there were no significant differences among experimental groups for chlorophyll *a*, chlorophyll *b* and total chlorophylls (*p* > 0.05). However, for total carotenoids, there were significant differences between animals fed Control diet and pigs fed *C. vulgaris* diets (*p* = 0.042), with approximately the double content of meat carotenoids in animals fed with the microalga. This could be explained by the much higher level of carotenoids in *C. vulgaris* diets that in the control diet (about nine times). In addition, there was also significant differences among groups fed with experimental diets for total chlorophylls and carotenoids (*p* = 0.038), being the sum two-fold higher in the group fed with CV + R diet compared with the control group; pork from animals fed with CV and CV + M diets had intermediate values of total pigments.

α-Tocopherol was the major diterpene in all groups fed with the experimental diets, while the other vitamin E homologues were present at lower concentrations. Concerning pigments, β-carotene (a pro-vitamin A) was not detected in pork, which could indicate that β-carotene in the diet is quickly metabolized into vitamin A [50], as animals cannot synthesize carotenoids by themselves [51]. *C. vulgaris*, due to the photosynthetic pathway, is also rich in pigments, such as chlorophylls and carotenoids. Despite the fact that β-carotene was not detected, the inclusion of 5% *C. vulgaris* in pig diets, combined or not with the two exogenous CAZyme mixtures, improved the carotenoid content of pork, thus providing further nutritional benefits for consumers. Total carotenoids were strongly in conformity with diet composition. Similar results were reported by Lemahieu et al. [52], who studied the effect of dietary supplementation of laying hens with different *n*-3 rich autotrophic microalgae, including *Chlorella*, on meat carotenoids. These authors reported that the transference of carotenoids from the microalgae to the meat provides an additional value for microalgae supplementation.

### 3.4. Total Lipids, Cholesterol and Fatty Acid Composition of Pork

Table 7 shows the effect of dietary inclusion of *C. vulgaris*, alone or combined with exogenous CAZymes, on total lipids, cholesterol, and fatty acid composition of longissimus lumborum muscle from pigs. Pork contents of total lipids and cholesterol were not affected by experimental diets (*p* > 0.05). In addition, experimental diets promoted only significant differences in the percentage of some minor fatty acids (<1% of total fatty acids). Control group had a higher percentage of the saturated fatty acid 10:0 relative to CV and CV + M groups (*p* = 0.013). In contrast, the percentages of the monounsaturated fatty acid 14:1*c*9 and *n*-3 fatty acids 18:3*n*-3, 18:4*n*-3, 20:3*n*-3, 20:5*n*-3, 22:5*n*-3, and 22:6*n*-3 were generally lower in the Control group relative to the *C. vulgaris* groups. Among microalga experimental groups, the group fed with CV + M usually had the highest percentage of these unsaturated fatty acids. In fact, both percentages of DPA and DHA increased 1.6-fold for CV + M diet comparing with the control diet. Contrarily, α-linolenic acid had higher percentages in all microalga-fed animals relative to the control group (+48%).

Regarding partial sums of fatty acids, total *n*-3 PUFA in pork had a significant increase of approximately 50% in microalga-fed groups comparing with the control group (*p* = 0.001). This increase reflects the individual effects of the predominant *n*-3 PUFA (α-linolenic acid, DPA, and DHA). The other partial sums of fatty acids, as well as the PUFA:SFA ratio, were not affected by the dietary treatment. However, the *n*-6:*n*-3 ratio decreased in all microalga-fed groups, in an extension of 24%, comparing with the Control group (*p* < 0.001). Feeding pigs with 5% of *C. vulgaris* increased the *n*-3 PUFA content in pork, which showed a correspondence between dietary and deposited *n*-3 PUFA in muscle. This finding reveals the ability of muscle to capture the precursor α-linolenic acid from *C. vulgaris* diets and its ability to convert it into *n*-3 PUFA derivatives. The *n*-3 long-chain PUFA (*n*-3 LC-PUFA), such as EPA and DHA, are of great interest for human diets due to their recognized positive effects, which includes anti-atherogenic, anti-thrombotic, and anti-inflammatory properties [53]. In fact, a well-balanced fatty acids intake is crucial to reduce the risk of cardiovascular and related diseases [54]. However, the intake of *n*-3 PUFA remains relatively low in human populations. In Europe, *n*-3 PUFA consumption is inferior to the recommendations of several international health organizations, which advise consuming one *n*-3 PUFA for five *n*-6 PUFA [55]. Although the intake of 250 mg per day already affords protection against cardiovascular diseases [56], the recommended daily intake of *n*-3 LC-PUFA ranges from 140 to 667 mg/day [57]. Herein, the dietary inclusion of 5% *C. vulgaris* in pig diets, supplemented or not with the two CAZyme mixtures, could be an interesting source to supply these beneficial fatty acids to animals and humans, since the usual *n*-3 PUFA content in pig muscle is very low (about 0.41–0.68 g/100 g of total fatty acids) [58]. In opposition to our findings, El-Bahr et al. [59] found higher levels of *n*-3 fatty acids, particularly of EPA and DHA, in breast muscle of broiler chickens fed *Spirulina platensis* and *Amphora coffeaformis* compared to those fed *C. vulgaris* and control birds. Interestingly, fatty acid profile in the microalgae supplemented contrasted with that of poultry meat, since *C. vulgaris* had higher *n*-3 fatty acids than *S. platensis* and *A. coffeaformis* [59].

Concerning the ratio of *n*-6:*n*-3 PUFA, pork from microalga-fed groups had lower values than that from Control group (−21%). Although these lower values are more health-protecting to the consumers, they are considerable higher (approximately 12) than the recommended ratio of 4 to prevent cardiovascular diseases [60].

### 3.5. Oxidative Stability of Pork

Table 8 displays the effect of experimental diets on oxidative stability of pig longissimus lumborum muscle after 0, 4, and 8 days of storage at 4 °C. Data showed that for 0 days of storage TBARS are not detected in any group fed with the different experimental diets, as well as for the group fed with CV + R diet with 4 days of storage. Although TBARS are detected for the other groups at day 4, and for all groups after 8 days of storage, experimental diets did not cause significant effects among them with regard to meat oxidative stability (*p* > 0.05). To complement these results, TBARS were also quantified after 0, 30, 120, and 300 min of chemical induction of lipid oxidation, through a ferrous/hydrogen peroxide system. No significant differences were observed among experimental diets for each time of lipid oxidation induction (*p* > 0.05), in spite of a significant increase of TBARS concentration between 0 and 30 min of lipid oxidation induction (*p* = 0.0001). Figure 1 presents the values of TBARS after 0, 30, 120, and 300 min of chemical induction of pork lipid oxidation for each experimental diet. The chemical lipid oxidation induced by the Fenton reaction corroborates the values of TBARS found for pork with the conventional TBARS method. In the current study, all the TBARS values during storage were below to the 0.9 mg malondialdehyde/kg of meat reported by Jayasingh and Cornforth [61] for ground and cooked pork meat. TBARS values above 0.5 mg malondialdehyde/kg of fresh meat are considered critical because at this level of lipid oxidation rancid odor and taste can be already detected by consumers [62]. Our values were all below this critical point, with exception of pork from CV + R group with eight days of storage, which was slightly higher (0.517).

The inclusion of microalgae rich in antioxidants as natural feed ingredients in animal diet can be a promising and sustainable alternative to enhance not only the nutritional value and health aspects of pork lipids, decreasing the ratio *n*-6:*n*-3 PUFA, but also delaying meat susceptibility to lipid oxidation [63]. However, PUFA represent the best candidates for the propagation of oxidative reactions that could depreciate the sensory and nutritional properties of foods [64]. Herein, the incorporation of 5% *C. vulgaris* in pig diets did not protect pork lipids from peroxidation, which is probably related to similar contents of PUFA, in spite of an important increase of carotenoids in microalga-fed groups in comparison to the control group. Baňoch et al. [46] and Vossen et al. [65] also documented no changes on pork and dry cured ham oxidative stability with the incorporation of 0.0002% and 0.3–1.2% levels of *Chlorella* sp., respectively. Notwithstanding *C. vulgaris* is an excellent source of antioxidant compounds, such as α-tocopherol and carotenoids, as previously documented by Safi et al. [66], the oxidative stability of pork did not reflect the antioxidant activity of *C. vulgaris*. In addition, Müller et al. [67] showed a large variation on the reactivity of the different types of carotenoids toward antioxidant activity. Therefore, changes in antioxidant activity are not only associated to the quantity of carotenoids but also with the specific characteristics of carotenoids identity [68]. This aspect deserves further investigation.

## 4. Conclusions

Dietary incorporation of 5% *C. vulgaris* did not negatively affect neither growth variables of finishing pigs nor carcass and meat quality traits (physicochemical and sensory analyses). In contrast, the inclusion of this microalga at this level in finishing pig diets improved the nutritional value of pork fat, through the increase of the beneficial lipid-soluble antioxidant pigments and *n*-3 PUFA, as well as the decrease of the *n*-6:*n*-3 ratio. In addition, the use of carbohydrases in the feed did not improve the digestive utilization of this microalga by pigs, at this incorporation level.

Overall, data indicate that *C. vulgaris* can be included in finishing pig diets up to 5%, with no need of feed enzymes supplementation, to increase pork fat nutritional value without impairing pig performance. As far as we know, this is the first study depicting the feasibility of the use of *C. vulgaris* as an alternative sustainable ingredient (incorporation at high levels) in swine feeding. In order to maximize both, the sustainability of swine diets and the pork nutritional quality, further research should be conducted with higher incorporation levels of *C. vulgaris*, combined or not with exogenous carbohydrases.

## Figures and Tables

**Figure 1 animals-10-02384-f001:**
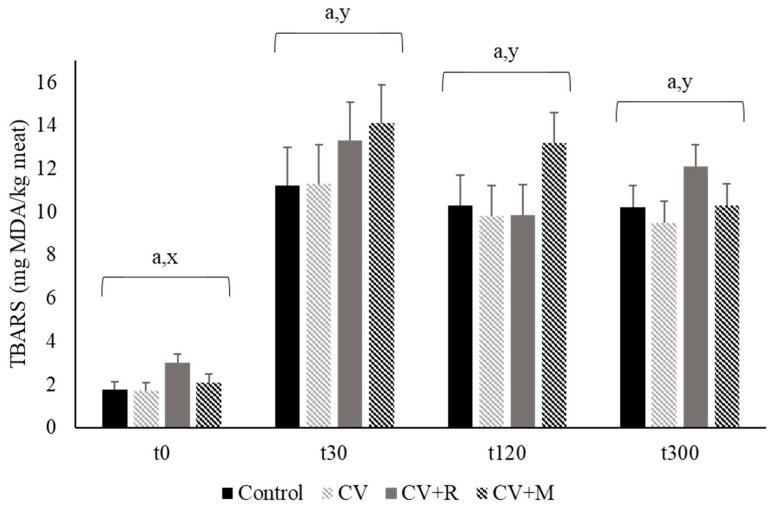
Determination of thiobarbituric acid reactive substances (TBARS) of pig longissimus lumborum muscle after time (t0, t30, t120 and t300 min) of chemical induction of lipid oxidation. Experimental diets: Control-corn-soybean basal diet; CV-basal diet plus 5% *C. vulgaris*; CV + R -basal diet plus 5% *C. vulgaris* + 0.005% Rovabio^®^ Excel AP; CV + M-basal diet plus 5% *C. vulgaris* + 0.01% mix of 4 CAZymes. Values with different letters within diet (a) and time (x,y) are significantly different (*p* < 0.05).

**Table 1 animals-10-02384-t001:** Ingredients and additives of the experimental diets (%, as fed basis).

Ingredients (%)	Experimental Diets			
	Control	CV	CV + R	CV + M
Corn	56	56	56	56
Soybean meal	19.3	11.7	11.6	11.7
Barley	10	10	10	10
Sunflower meal	5.4	6.8	6.8	6.8
Wheat	5	5	5	5
Calcium carbonate	1.3	1.2	1.2	1.2
Soybean oil	0.5	0.2	0.2	0.2
Wheat bran	0.4	1.7	1.66	1.65
Salt	0.4	0.4	0.4	0.4
Vitamin–trace mineral premix ^1^	0.4	0.4	0.4	0.4
Dicalcium phosphate	0.26	0.36	0.36	0.36
Sodium bicarbonate	0.1	0.1	0.1	0.1
Betaine-HCl	0.15	0.15	0.15	0.15
Mold inhibitor mixture ^2^	0.075	0.075	0.075	0.075
Fatty acid mixture ^3^	0.075	0.075	0.075	0.075
L-Lysine	0.41	0.57	0.57	0.57
L-Threonine	0.1180	0.2000	0.2000	0.2000
DL-Methionine	0.0712	0.1080	0.1080	0.1080
L-Tryptophan	0.0064	-	-	-
*Chlorella vulgaris*	-	5	5	5
Mix of 4 CAZymes	-	-	-	0.01
Rovabio^®^ Excel AP	-	-	0.005	-

Experimental diets: Control-corn-soybean basal diet; CV-basal diet plus 5% *C. vulgaris*; CV + R -basal diet plus 5% *C. vulgaris* + 0.005% Rovabio^®^ Excel AP; CV + M-basal diet plus 5% *C. vulgaris* + 0.01% mix of 4 CAZymes. ^1^ VitaTec (Tecadi, Santarém, Portugal). Provided the following nutrients per kg of diet: Premix provided per kg of complete diet: 6000 IU vitamin A; 1500 IU vitamin D_3_; 15 mg vitamin E (acetate _DL_-alpha-tocopherol); 0.3 mg vitamin B_2_; 3.75 mg vitamin B_12_; 0.1 mg biotin; 12 mg calcium pantothenate, 15 mg nicotinic acid; 0.75 mg folic acid; 200 mg choline chloride; 15 mg Cu (cupric sulphate pentahydrate); 100 mg Zn (zinc oxide); 35 mg Mn (manganese oxide); 0.7 mg I (potassium iodide); 0.05 mg Co (basic cobaltous carbonate mono hydrous); 0.2 mg Se (sodium selenite); 80 mg Fe(ferrous carbonate); and 0.2 mg butylated hydroxyl-toluene. ^2^ Yeast extracts, high absorbent clay mineral, plant derivatives, calcium propionate and antioxidant premix (Escent^®^ S, Innovad, Berchem, Belgium). ^3^ Esterified butyric acid, medium chain fatty acid, plant extract and essential oil (Lumance^®^, Innovad, Berchem, Belgium).

**Table 2 animals-10-02384-t002:** Chemical composition of *Chlorella vulgaris* and experimental diets.

Item	Microalga *C. vulgaris*	Experimental Diets			
		Control	CV	CV + R	CV + M
ME, kcal/kg DM ^1^	3557	3576	3540	3644	3547
**Proximate composition, %**					
Dry matter	93.1	90.0	89.7	89.5	90.0
Crude protein	42.8	14.0	15.9	15.2	15.2
Starch	1.86	45.5	45.3	44.7	47.4
Crude fat	8.73	2.60	3.00	3.10	3.10
Crude fiber	1.52	4.60	5.00	5.30	5.20
NDF	1.05	13.7	13.9	12.7	13.7
ADF	0.286	4.90	5.50	5.50	5.90
Ash	11.8	4.03	4.70	4.60	4.60
**Amino acid composition, %**					
Alanine	2.77	0.682	0.848	0.806	0.776
Arginine	3.89	0.890	1.11	1.03	0.969
Asparagine	0.062	0.023	0.022	0.015	0.018
Aspartate	3.04	1.00	1.08	1.11	1.01
Cysteine	0.665	0.292	0.268	0.237	0.248
Glutamate	4.07	2.33	2.22	2.21	2.10
Glutamine	0.016	nd	nd	nd	nd
Glycine	1.72	0.544	0.687	0.614	0.584
Histidine	0.654	0.512	0.593	0.528	0.489
Hydroxyproline	0.741	0.880	1.33	1.19	1.16
Isoleucine	1.26	0.478	0.536	0.521	0.482
Leucine	2.45	0.942	1.05	1.03	0.984
Lysine	2.63	1.04	1.43	1.42	1.32
Methionine	0.451	0.116	0.124	0.144	0.088
Phenylalanine	1.49	0.578	0.634	0.621	0.587
Proline	1.87	1.06	1.04	1.04	1.01
Serine	1.56	0.689	0.771	0.727	0.679
Threonine	2.32	0.761	0.989	1.00	0.943
Tryptophan	0.471	0.156	0.172	0.147	0.133
Tyrosine	1.18	0.429	0.495	0.470	0.437
Valine	3.52	1.20	1.43	1.32	1.26
**Fatty acid profile, % total fatty acids**					
14:0	1.13	0.150	0.218	0.190	0.190
16:0	17.2	16.3	16.6	16.3	16.5
16:1*c*9	3.90	0.228	1.14	0.989	0.972
17:0	0.234	0.189	0.182	0.153	0.154
17:1*c*9	0.610	0.038	0.704	0.739	0.732
18:0	3.00	2.89	3.29	3.11	3.08
18:1*c*9	11.7	27.4	27.4	27.6	27.5
18:1*c*11	nd	0.885	1.70	1.38	1.42
18:2*n*-6	11.2	48.1	44.1	45.1	44.9
18:3*n*-3	10.1	2.57	3.47	3.28	3.28
20:0	0.174	0.528	0.513	0.517	0.500
20:1*c*11	0.127	0.292	0.288	0.320	0.320
20:5*n*-3	nd	nd	nd	nd	nd
22:0	0.060	0.304	0.294	0.262	0.266
22:1*n*-9	nd	0.155	0.155	0.131	0.149
22:6*n*-3	nd	nd	nd	nd	nd
**Diterpene profile, μg/g**					
α-Tocopherol	19.2	16.5	18.7	19.4	16.5
α-Tocotrienol	nd	4.84	3.70	3.88	4.36
β-Tocopherol	0.340	0.380	0.268	0.244	0.258
γ -Tocopherol	0.520	3.53	2.74	2.35	2.65
γ-Tocotrienol	0.560	7.23	5.93	7.30	6.02
δ-Tocopherol	0.360	0.340	0.331	0.312	0.314
δ-Tocotrienol	nd	0.287	0.230	0.246	0.247
**Pigments, μg/g**					
β-Carotene	198	1.19	7.10	7.40	6.49
Chlorophyll *a* ^2^	906	4.31	127	139	126
Chlorophyll *b* ^3^	171	7.46	33.9	36.6	34.2
Total Chlorophylls ^4^	1077	11.8	161	176	160
Total Carotenoids ^5^	228	3.97	36.5	39.5	34.9
Total Chlorophylls+ carotenoids ^6^	1305	15.7	198	215	195

Experimental diets: Control-corn-soybean basal diet; CV-basal diet plus 5% *C. vulgaris*; CV + R -basal diet plus 5% *C. vulgaris* + 0.005% Rovabio^®^ Excel AP; CV + M-basal diet plus 5% *C. vulgaris* + 0.01% mix of 4 CAZymes. ME–metabolized energy; DM–dry matter; NDF–neutral detergent fiber; ADF–acid detergent fiber; nd–not detected. ^1^ Metabolizable energy (kcal/kg DM) = 4412−11.06 × Ash (g/kg DM) + 3.37 × Crude Fat (g/kg DM) − 5.18 × ADF (g/kg DM). ^2^ Chlorophyll *a* = 11.24 × A662 nm − 2.04 × A645 nm. ^3^ Chlorophyll *b* = 20.13 × A645 nm − 4.19 × A662 nm. ^4^ Total chlorophylls (Ca + b) = 7.05 × A662 nm + 18.09 × A645 nm. ^5^ Total carotenoids (Cx + c) = (1000 × A470 nm − 1.90 ×Ca − 63.14 × Cb)/214. ^6^ Total chlorophylls and carotenoids = (Ca + b) + (Cx + c).

**Table 3 animals-10-02384-t003:** Effect of experimental diets on growth performance and carcass characteristics of pigs.

Item	Control	CV	CV + R	CV + M	SEM	*p*-Value
**Growth performance**						
Initial weight, kg	62.8	56.1	58.4	59.4	1.79	0.075
Final weight, kg	101	101	101	101	0.643	0.927
ADFI, kg	2.56	2.67	2.65	2.60	0.052	0.409
ADG, kg	0.959	1.08	1.01	1.04	0.037	0.141
FCR	2.69	2.49	2.63	2.55	0.079	0.286
G:F, kg/kg	0.374	0.404	0.382	0.398	0.011	0.244
**Carcass characteristics**						
HCW, kg	80.1	79.5	79.3	78.9	0.735	0.703
Carcass yield, %	77.4	77.1	76.9	76.8	0.430	0.749
Perirenal fat, kg	0.897 ^b^	0.666 ^a^	0.806 ^ab^	0.711 ^ab^	0.055	0.026
Mesenteric fat, kg	0.525	0.530	0.572	0.583	0.024	0.231
P_2_ backfat thickness, mm	6.38	5.54	7.17	6.40	0.633	0.359
L6 backfat thickness, mm	9.33	10.1	10.8	9.64	0.758	0.535
S2 backfat thickness, mm	4.98	5.22	5.42	5.77	0.737	0.891
Loin weight, kg	2.14	2.11	2.10	2.18	0.066	0.850
Drip loss % ^1^	5.82	5.63	7.27	6.51	0.460	0.065

Experimental diets: Control-corn-soybean basal diet; CV-basal diet plus 5% *C. vulgaris*; CV + R -basal diet plus 5% *C. vulgaris* + 0.005% Rovabio^®^ Excel AP; CV + M-basal diet plus 5% *C. vulgaris* + 0.01% mix of 4 CAZymes. SEM–standard error of the mean; ADFI–average daily feed intake; ADG – average daily weight gain; FCR–feed conversion ratio; G:F–gain-feed ratio; HCW–hot carcass weight; P_2_–at the last rib position; L6–at the last lumbar vertebra; S2–at the second sacral vertebra. ^1^ Measured as the amount of drip between 24 h and 144 h post mortem. ^a, b^ Values within a row with different superscripts differ significantly at *p* < 0.05.

**Table 4 animals-10-02384-t004:** Effect of experimental diets on meat quality traits of longissimus lumborum muscle from pigs.

Item	Control	CV	CV + R	CV + M	SEM	*p*-Value
**Temperature, °C**						
45 min	39.9	39.8	39.7	40.0	0.246	0.911
pH						
45 min	6.11	6.34	6.12	6.28	0.109	0.351
24 h	5.49	5.54	5.50	5.51	0.016	0.260
**Color measurements**						
Lightness (L*)	57.0	56.5	57.9	56.9	0.976	0.791
Redness (a*)	6.50	5.68	6.28	6.39	0.600	0.770
Yellowness (b*)	7.26	6.46	7.24	7.07	0.526	0.679
**Other traits**						
WBSF, kg	6.92	7.17	6.44	6.95	0.373	0.574
Cooking loss, %	30.8	30.7	31.0	30.1	0.605	0.740

Experimental diets: Control-corn-soybean basal diet; CV-basal diet plus 5% *C. vulgaris*; CV + R -basal diet plus 5% *C. vulgaris* + 0.005% Rovabio^®^ Excel AP; CV + M-basal diet plus 5% *C. vulgaris* + 0.01% mix of 4 CAZymes. SEM–standard error of the mean. * Color parameters CIE L* a* b* system. WBSF–Warner-Bratzler shear force.

**Table 5 animals-10-02384-t005:** Effect of experimental diets on sensory panel scores of longissimus lumborum muscle from pigs.

Item	Control	CV	CV + R	CV + M	SEM	*p*-Value
Tenderness	4.45	4.61	4.57	4.54	0.117	0.788
Juiciness	3.72	3.85	3.74	3.84	0.111	0.760
Flavor	4.09	4.20	4.29	4.20	0.109	0.649
Off-flavor	0.061	0.111	0.171	0.131	0.029	0.064
Flavor acceptability	5.55	5.29	5.36	5.32	0.104	0.260
Overall acceptability	5.23	5.22	5.13	5.10	0.101	0.756

Experimental diets: Control-corn-soybean basal diet; CV-basal diet plus 5% *C. vulgaris*; CV + R -basal diet plus 5% *C. vulgaris* + 0.005% Rovabio^®^ Excel AP; CV + M-basal diet plus 5% *C. vulgaris* + 0.01% mix of 4 CAZymes. SEM–standard error of the mean.

**Table 6 animals-10-02384-t006:** Effect of experimental diets on vitamin E profile and pigments of longissimus lumborum muscle from pigs.

Item	Control	CV	CV + R	CV + M	SEM	*p*-Value
**Diterpene profile, µg/100 g**						
α-Tocopherol	95.4	73.6	74.9	79.4	6.2	0.062
γ-Tocopherol	3.5	3.7	3.5	3.2	0.2	0.441
γ-Tocotrienol	10.2	9.0	10.4	8.2	1.9	0.821
**Pigments, µg/100 g**						
β-Carotene	nd	nd	nd	nd	-	-
Chlorophyll a	14.7	23.9	31.3	28.0	4.75	0.094
Chlorophyll b	27.7	47.2	56.9	54.7	9.00	0.109
Total chlorophylls	42.4	71.2	88.1	82.8	13.7	0.103
Total carotenoids	7.18 ^a^	16.4 ^b^	16.4 ^b^	15.1 ^b^	2.55	0.042
Total chlorophylls and carotenoids	49.6 ^a^	87.6 ^ab^	104 ^b^	97.9 ^ab^	13.9	0.038

Experimental diets: Control-corn-soybean basal diet; CV-basal diet plus 5% *C. vulgaris*; CV + R -basal diet plus 5% *C. vulgaris* + 0.005% Rovabio^®^ Excel AP; CV + M-basal diet plus 5% *C. vulgaris* + 0.01% mix of 4 CAZymes. SEM–standard error of the mean. nd–not detected. ^a, b^ Values within a row with different superscripts differ significantly at *p* < 0.05.

**Table 7 animals-10-02384-t007:** Effect of experimental diets on total lipid content, total cholesterol and fatty acid (FA) composition of longissimus lumborum muscle from pigs.

Item	Control	CV	CV + R	CV + M	SEM	*p*-Value
Total lipids, g/100 g	1.18	1.03	1.05	0.933	0.073	0.141
Cholesterol, mg/g	0.363	0.363	0.361	0.367	0.015	0.993
**FA composition, g/100 g FA**						
10:0	0.053 ^b^	0.023 ^a^	0.042 ^ab^	0.023 ^a^	0.007	0.013
12:0	0.056	0.045	0.053	0.051	0.006	0.536
14:0	1.05	0.952	0.994	0.904	0.045	0.126
14:1*c*9	0.034 ^a^	0.062 ^ab^	0.064 ^ab^	0.068 ^b^	0.008	0.021
15:0	0.081	0.072	0.067	0.069	0.007	0.519
DMA 16:0	0.089	0.047	0.054	0.140	0.029	0.107
16:0	23.4	22.8	23.2	22.5	0.279	0.119
16:1*c*7	0.335	0.352	0.338	0.388	0.015	0.065
16:1*c*9	2.94	2.67	2.79	2.42	0.131	0.054
17:0	0.432	0.435	0.417	0.460	0.038	0.882
17:1*c*9	0.340	0.369	0.363	0.334	0.023	0.647
DMA 18:0	0.045	0.019	0.067	0.076	0.032	0.597
DMA 18:1	0.023	0.006	0.034	0.039	0.020	0.637
18:0	11.9	11.6	11.9	12.2	0.373	0.698
18:1*c*9	37.3	36.1	36.7	34.8	0.933	0.270
18:1*c*11	3.99	3.94	3.91	3.79	0.072	0.260
18:2*n*-6	11.8	13.4	12.4	13.9	0.846	0.291
18:2*t*9*t*12	0.039	0.034	0.026	0.032	0.006	0.494
18:3*n*-6	0.121	0.129	0.123	0.133	0.014	0.934
18:3*n*-3	0.279 ^a^	0.408 ^b^	0.377 ^b^	0.381 ^b^	0.020	>0.001
18:4*n*-3	0.027 ^a^	0.050 ^b^	0.041 ^ab^	0.058 ^b^	0.006	0.004
20:0	0.167	0.154	0.161	0.171	0.007	0.422
20:1*c*11	0.604	0.593	0.594	0.595	0.033	0.996
20:2*n*-6	0.341	0.358	0.326	0.336	0.018	0.675
20:3*n*-6	0.362	0.415	0.383	0.457	0.035	0.270
20:4*n*-6	2.30	2.72	2.40	2.93	0.280	0.368
20:3*n*-3	0.056 ^a^	0.080 ^ab^	0.089 ^b^	0.092 ^b^	0.008	0.008
20:5*n*-3	0.064 ^a^	0.119 ^b^	0.114 ^b^	0.112 ^b^	0.015	0.042
22:0	0.068	0.070	0.069	0.088	0.008	0.240
22:1*n*-9	0.047	0.049	0.055	0.043	0.008	0.740
22:5*n*-3	0.266 ^a^	0.385 ^ab^	0.356 ^ab^	0.428 ^b^	0.040	0.036
22:6*n*-3	0.241 ^a^	0.328 ^ab^	0.342 ^ab^	0.393 ^b^	0.038	0.035
23:0	0.162	0.189	0.170	0.211	0.021	0.366
Others	0.946	1.03	1.02	1.35	0.227	0.615
**Partial sums of FA, g/100 g FA**						
SFA	37.4	36.4	37.1	36.7	0.534	0.564
MUFA	45.6	44.2	44.8	42.4	1.09	0.213
PUFA	15.9	18.4	17.0	19.3	1.25	0.243
*n*-6 PUFA	14.9	17.0	15.6	17.8	1.17	0.306
*n*-3 PUFA	0.932 ^a^	1.37 ^b^	1.32 ^b^	1.46 ^b^	0.093	0.001
**Ratios of FA**						
PUFA:SFA	0.427	0.508	0.461	0.530	0.038	0.232
*n*-6:*n*-3	16.1 ^b^	12.3 ^a^	11.9 ^a^	12.3 ^a^	0.395	<0.001

Experimental diets: Control-corn-soybean basal diet; CV-basal diet plus 5% *C. vulgaris*; CV + R -basal diet plus 5% *C. vulgaris* + 0.005% Rovabio^®^ Excel AP; CV + M - basal diet plus 5% *C. vulgaris* + 0.01% mix of 4 CAZymes. SEM–standard error of the mean; FA–fatty acids; DMA–dimethylacetal; SFA–saturated fatty acids; MUFA–monounsaturated fatty acids; PUFA-polyunsaturated fatty acids. SFA = Sum of (10:0, 12:0, 14:0, 15:0, 16:0, 17:0, 18:0, 20:0, 22:0 and 23:0). MUFA = Sum of (14:1*c*9, 16:1*c*7, 16:1*c*9, 17:1*c*9, 18:1*c*9, 18:1*c*11, 20:1*c*11 and 22:1*n*-9). PUFA = Sum of (18:2*n*-6, 18:3*n*-6, 18:2*t*9*t*12, 18:3*n*-3, 18:4*n*-3, 20:2*n*-6, 20:3*n*-6, 20:4*n*-6, 20:3*n*-3, 20:5*n*-3, 22:5*n*-3 and 22:6*n*-3). *n*-3 PUFA = Sum of (18:3*n*-3, 18:4*n*-3, 20:3*n*-3, 20:5*n*-3, 22:5*n*-3 and 22:6*n*-3). *n*-6 PUFA = Sum of (18:2*n*-6, 18:3*n*-6, 20:2*n*-6, 20:3*n*-6, and 20:4*n*-6). ^a, b^ Values within a row with different superscripts differ significantly at *p* < 0.05.

**Table 8 animals-10-02384-t008:** Effect of experimental diets on oxidative stability of pig longissimus lumborum muscle after 0, 4 and 8 days of storage at 4 °C.

TBARS, mg MDA/kg meat	Control	CV	CV + R	CV + M	SEM	*p*-Value
Day 0	nd	nd	nd	nd	-	-
Day 4	0.027	0.047	nd	0.031	0.017	0.604
Day 8	0.186	0.174	0.517	0.160	0.142	0.234

Experimental diets: Control-corn-soybean basal diet; CV-basal diet plus 5% *C. vulgaris*; CV + R -basal diet plus 5% *C. vulgaris* + 0.005% Rovabio^®^ Excel AP; CV + M-basal diet plus 5% *C. vulgaris* + 0.01% mix of 4 CAZymes. TBARS–Thiobarbituric acid reactive substances; MDA–malonaldehyde; SEM–standard error of the mean; nd–not detected.

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
