# Peer review of "A High Dietary Incorporation Level of Chlorella vulgaris Improves the Nutritional Value of Pork Fat without Impairing the Performance of Finishing Pigs"

_animals, 2020, doi:10.3390/ani10122384_

Round 1
Reviewer 1 Report
L-120- The distribution of animals is not clear to me: in each pen 1 pig; each pig a type of diet...
Why not 4 pens with 10 pigs, isolated and all in the same conditions and eating each pen the same thing?
Or in any case, detail why they decided to have 10 isolated control animals, as well as the remaining groups. What were the advantages you thought you would obtain in this way? Please clarify this confusing point
L-157.How long was the experiment?
L-284. Even if they use each pen in each study group, the homogeneous conditions are not clear to me; 4 pigs eat different diets in the same space...Then the authors summarize the results of each individual from each pen/lot...I find this point confusing.
In general terms, it is serious work, highly relevant, and can serve as a model for future work in the area of animal supplementation with products of natural origin. The only point that confuses me, and I hope the authors will clarify, is the distribution of animals in groups.
Author Response
Reviewer #1: Comments and Suggestions for Authors:
- MATERIALS AND METHODS:
2.2. Animal Care, Experimental Design and Experimental Diets - L-120- The distribution
of animals is not clear to me: in each pen 1 pig; each pig a type of diet...
Why not 4 pens with 10 pigs, isolated and all in the same conditions and eating each pen the same thing? Or in any case, detail why they decided to have 10 isolated control animals, as well as the remaining groups. What were the advantages you thought you would obtain in this way? Please clarify this confusing point.
Reply: The 40 pigs were randomly housed in 10 pens, with 4 animals in each, and each animal in each pen was randomly assigned to one of the 4 experimental diets. This animal distribution enables a complete random of animals and dietary treatments, thus avoiding possible effects of pen location and animal interaction inside the pen. However, the animals were fed individually using a system of gates that isolated the pig while they were eating. Feed offered and refusals were recorded daily for each pig in order to obtain individual feed intake. This aspect was clarified in the manuscript (lines 121-123).
2.3. Animal Performance, Slaughter and Sampling - L-157 - How long was the experiment?
Reply: At the beginning of the trial the animals had the initial weight of 59.1 ± 5.69 kg (118 ± 8.1 days), and at the end of the trial they had the final weight of 101 ± 1.9 kg (159 ± 9.1 days). Therefore, the trial lasted for 41 ± 7.8 days. This information was included in the manuscript (line 166).
2.12. Statistical analysis - L-284. Even if they use each pen in each study group, the homogeneous conditions are not clear to me; 4 pigs eat different diets in the same space...Then the authors summarize the results of each individual from each pen/lot...I find this point confusing.
Reply: The 40 pigs were randomly housed in 10 pens, with 4 animals in each, and each animal in each pen was randomly assigned to one of the 4 different experimental diets. This animal distribution enables a complete random of animals and dietary treatments, thus avoiding possible effects of pen location and animal interaction inside the pen. However, the animals were fed individually using a system of gates that isolated the pig while they were eating. Feed offered and refusals were recorded daily for each pig in order to obtain individual feed intake. Therefore, the pig is the experimental unit. This aspect was clarified in the manuscript (lines 121-123).
In general terms, it is serious work, highly relevant, and can serve as a model for future work in the area of animal supplementation with products of natural origin. The only point that confuses me, and I hope the authors will clarify, is the distribution of animals in groups.
Reviewer 2 Report
The manuscript presented for review concerns still current issues related to the possibility of enrichment of animal products with ingredients beneficial to consumer health. In my opinion, it is a valuable contribution to science.
Some minor concerns are preented below:
- I think the authors should mention in the introduction part about fatty acids profile of C. vulgaris used in the study, especially in terms of n-3 acids group.
- I have some doubts regarding animals feeding (lines 117-120)- how was it possible to control the individual feed intake if the pigs were kept in collective pens and had ad libitum access to feed? I see there were four feeders in each pen, but how did you prevent using various feeders by particular animals?
- There were 40 animals, allocated to 10 pens, does it mean that each animal in particular pens was fed different diet?
- Table 2 - the uthors mentioned in the introductin part about the significance of EPA and DHA acids, why were they not analyzed in diet offered to pigs?
Author Response
Reviewer #2: Comments and Suggestions for Authors
The manuscript presented for review concerns still current issues related to the possibility of enrichment of animal products with ingredients beneficial to consumer health. In my opinion, it is a valuable contribution to science.
Some minor concerns are presented below:
I think the authors should mention in the introduction part about fatty acids profile of C. vulgaris used in the study, especially in terms of n-3 acids group.
Reply: The authors acknowledge the reviewer’s suggestion. A brief review about C. vulgaris fatty acid profile, including the important n-3 polyunsaturated fatty acids, was added to the introduction (lines 73 to 77). In fact, the levels of EPA (20:5n-3) and DHA (22:6n-3) in C. vulgaris and other Chlorella species are classically low to undetectable (see also Matos et al., 2020, International Journal of Food Science and Technology, 55, 303–312; Jay et al., 2018, IOP Conf. Ser.: Earth Environ. Sci., 141, 012015; Ohse et al., 2015, IDESIA, 33, 93–101; Zhang et al., 2014, Ann Microbiol, 64, 1239–1246). Only a few studies reported low amounts of these n-3 PUFA in C. vulgaris (Ferreira et al., 2017 International Food Research Journal, 24, 284–291; Rismani and Shariati, 2017 Brazilian Archives of Biology and Technology, 60, e17160555). Nevertheless, the fatty acid profile of microalgae is largely a function of cultivation conditions and growth phase at the harvest time, as is the case of the C. vulgaris (Ferreira et al., 2017; Rismani and Shariati, 2017) and other Chlorella species (Ferreira et al., 2017).
I have some doubts regarding animals feeding (lines 117-120) - how was it possible to control the individual feed intake if the pigs were kept in collective pens and had ad libitum access to feed? I see there were four feeders in each pen, but how did you prevent using various feeders by particular animals?
Reply: The 40 pigs were randomly housed in 10 pens, with 4 animals in each, and each animal in each pen was randomly assigned to one of the 4 different experimental diets. This animal distribution enables a complete random of animals and dietary treatments, thus avoiding possible effects of pen location and animal interaction inside the pen. However, the animals were fed individually using a system of gates that isolated the pig while they were eating. Feed offered and refusals were recorded daily for each pig in order to obtain individual feed intake. This aspect was clarified in the manuscript (lines 121-123).
There were 40 animals, allocated to 10 pens, does it mean that each animal in particular pens was fed different diet?
Reply: Each of the 4 animals in each pen was randomly assigned to one of the 4 different experimental diets. The animals were fed individually using a system of gates that isolated the pig while they were eating. Feed offered and refusals were recorded daily for each pig in order to obtain individual feed intake. This aspect was clarified in the manuscript (lines 121-123).
Table 2 - the authors mentioned in the introduction part about the significance of EPA and DHA acids, why were they not analyzed in diet offered to pigs?
Reply: As described in the Material and Methods section, the detailed fatty acid profile of C. vulgaris and the experimental diets offered to pigs were analyzed. However, EPA and DHA levels in both samples were undetectable by GC analysis. In order to clarify it, EPA (20:5n-3) and DHA (22:6n-3) fatty acids are now presented in Table 2 as not detected (nd) (see Table 2).